# OpenSage: Self-programming Agent Generation Engine

Hongwei Li [* 1]   Zhun Wang [* 2]   Qinrun Dai [3]   Yuzhou Nie [1]   Jinjun Peng [4]   Ruitong Liu [3]   Jingyang Zhang [5]
Kaijie Zhu [1]   Jingxuan He [2]   Lun Wang [6]   Yangruibo Ding [7]   Yueqi Chen [3]   Wenbo Guo [1]   Dawn Song [2]

## Abstract

Agent development kits (ADKs) provide effective platforms and tooling for constructing agents, and their designs are critical to the constructed agents' performance, especially the functionality for agent topology, tools, and memory. However, current ADKs either lack sufficient functional support or rely on humans to manually design these components, limiting agents' generalizability and overall performance. We propose OpenSage, the first ADK that enables LLMs to automatically create agents with self-generated topology and toolsets while providing comprehensive and structured memory support. OpenSage offers effective functionality for agents to create and manage their own sub-agents and toolkits. It also features a hierarchical, graph-based memory system for efficient management and a specialized toolkit tailored to software engineering tasks. Extensive experiments across four state-of-the-art benchmarks with various backbone models demonstrate the advantages of OpenSage over existing ADKs. We also conduct rigorous ablation studies to demonstrate the effectiveness of our design for each component. We believe OpenSage can pave the way for the next generation of agent development, shifting the focus from human-centered to AI-centered paradigms. Our code is available at `https://github.com/opensage-agent/opensage-adk`.

## 1. Introduction

AI agents are under explosive growth, driven by their promising performance across diverse application domains (Wang

*Equal contribution   [1]UC Santa Barbara [2]UC Berkeley [3]University of Colorado Boulder [4]Columbia University [5]Duke University [6]Google DeepMind [7]UC Los Angeles. Correspondence to: Hongwei Li <hongwei@ucsb.edu>, Zhun Wang <zhun.wang@berkeley.edu>.

*Proceedings of the 43rd International Conference on Machine Learning*, Seoul, South Korea. PMLR 306, 2026. Copyright 2026 by the author(s).

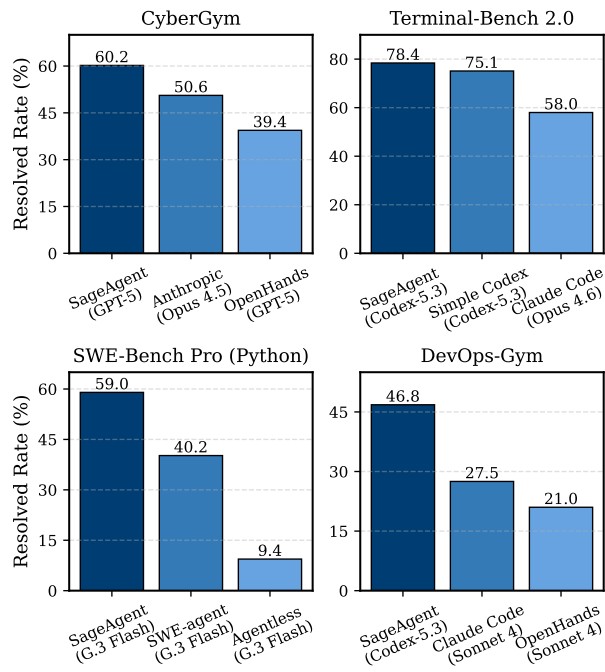

*Figure 1.* SageAgent (via OpenSage) vs. SOTA agents and ADKs on four popular agentic benchmarks. "G.3" refers to Gemini 3.

et al., 2025a; Novikov et al., 2025; Ghafarollahi & Buehler, 2025; Ramos et al., 2025; Chu et al., 2025; Xi et al., 2025; Tang et al., 2026). This rapid evolution necessitates effective frameworks for agent construction. As such, both academia and industry have developed Agent Development Kits (ADKs) to provide the infrastructure and functionality required to construct agents equipped with complex tools and memory. A well-designed ADK is critical to both the utility and performance of the resulting agents, which are largely determined by three core architectural components: **agent topology, tooling system, and memory system**.

First, agent topology defines the agentic system's structure, including the architecture and tasks of individual agents, as well as their dependencies and interaction mechanisms. It forms the core of agents and directly determines their capability and effectiveness (Zhou et al., 2025; Qian et al., 2024). Second, agents interact with the environment through tools. A restricted toolset limits agents' retrieved informa-

tion, frequently leading to hallucinations. It also constrains the agents' action spaces and the range of tasks they can accomplish (Li, 2025). Finally, memory systems enable agents to learn from past interactions, which is particularly helpful when executing complex tasks (Behrouz et al., 2025; Yan et al., 2025; Zhang et al., 2025; Suzgun et al., 2025).

SOTA ADKs, including Google ADK (Google, 2026), OpenHands SDK (Wang et al., 2024), OpenAI ADK (OpenAI, 2026), Claude ADK (Anthropic, 2026b), and LangChain (Inc, 2026), provide basic functionality but still rely on humans to design these three core components. As a result, the required substantial human effort and domain expertise limit the scalability of agent construction, while the lack of dynamic adaptation in agent structure and toolsets across tasks constrains generalizability. This human-centered paradigm resembles early-stage machine learning that relied on tedious, handcrafted features. Modern ML, however, requires only a base model architecture and learns capable models directly from experience and feedback signals (denoted AI-centered paradigm).

To enable an AI-centered paradigm for agent construction, we propose OpenSage (**Open S**elf-programming **A**gent **G**eneration **E**ngine), the next-generation ADK that allows AI to create agent systems, construct tools, and manage memory for context storage and retrieval. ❶ Technically, OpenSage supports dynamic creation, execution, and termination of sub-agents during task execution (Section 3.2). This mechanism enables various agent topologies based on different tasks, where two types are most commonly seen: 1) vertical agent topology, which decomposes complex tasks into sequential sub-tasks to be completed by specialized sub-agents, and 2) horizontal agent topology, where multiple sub-agents simultaneously execute the same task using distinct plans, with their results then integrated through an agent ensemble mechanism. ❷ Beyond topological flexibility, OpenSage empowers AI to construct its own tools for targeting tasks and provides tool management, including overall tool orchestration, execution isolation, and state management (Section 3.3). OpenSage also integrates a domain-specific toolkit optimized for software engineering tasks, which would be infeasible with existing ADKs, since they cannot support heterogeneous tools that require different execution environments. ❸ Finally, OpenSage designs a hierarchical memory system that combines short-term history with long-term system knowledge, managed by a dedicated memory agent (Section 3.4).

We evaluate OpenSage on four SOTA benchmarks, including Terminal-Bench 2.0 (The Terminal-Bench Team, 2025), CyberGym (Wang et al., 2025c), DevOps-Gym (Tang et al., 2026) and SWE-Bench Pro (Deng et al., 2025), using various backbone models (Section 4.1). Our results show that the agents constructed by OpenSage significantly outper-

*Table 1.* Comparison between OpenSage and SOTA ADKs in key features. ● means full support; ◐ means partial or limited support; ○ means not supported.

| Category | Feature | OpenSage | Google | OpenAI | Claude | OpenHands | LangChain |
|---|---|---|---|---|---|---|---|
| **Topology** | AI-created topology | ● | ○ | ○ | ○ | ○ | ○ |
| | Agent management | ● | ◐ | ◐ | ◐ | ◐ | ◐ |
| | Agent ensemble | ● | ○ | ○ | ○ | ○ | ○ |
| **Tool** | AI-written tools | ● | ○ | ○ | ◐ | ○ | ○ |
| | Tool management | ● | ○ | ○ | ◐ | ◐ | ○ |
| **Memory** | AI-created memory | ● | ○ | ○ | ○ | ○ | ○ |
| | Graph-based structure | ● | ○ | ○ | ○ | ○ | ○ |
| | AI-driven management | ● | ○ | ○ | ○ | ○ | ○ |

form the SOTA ADKs on all benchmarks (Figure 1). Further, to validate the agent topology design, we assess the effectiveness of both vertical and horizontal topologies, as well as using this feature to enhance cost-efficiency (Section 4.2). We also validate the efficiency of our tooling system (Section 4.3) and memory system designs (Section 4.4) through ablation studies. The substantial performance gap between OpenSage agents with and without these functionalities demonstrates their necessity.

Throughout our experiments, we observe the backbone model actively creating sub-agents for different sub-tasks, together with tailored system prompts that the agent synthesizes on its own. It also autonomously creates specialized sub-agents to manage toolsets with related functionalities, such as dedicated debugging agents. Notably, by leveraging its tool-writing capabilities, the model constructs task-specific tools rather than relying solely on the general-purpose tools provided initially. Furthermore, our hierarchical memory system and dedicated memory agent significantly optimized context length while preventing redundant and repeated queries. All these behaviors contribute to a substantial improvement in agent performance. However, we also noted that SOTA models do not yet utilize these advanced features consistently or may misuse them, indicating a gap in capability. These results show that while such features are highly promising and effective, we need to develop stronger AI models to fully realize their potential. To the best of our knowledge, OpenSage is the first ADK to pioneer an AI-centered paradigm for agent construction. It establishes the foundation for unleashing AI's potential of self-evolving agents.

**Conflict of Interest Disclosure.** The author L.W. is employed by Google DeepMind, which leads the development of the Gemini model family, which was among the ones evaluated in this paper.

## 2. Existing ADKs and Limitations

Table 1 compares OpenSage with SOTA ADKs across agent topology, tooling system, and memory system.

**Topology.** No existing ADKs (Wang et al., 2024; Google, 2026; OpenAI, 2026; Anthropic, 2026b; Wang et al., 2024; Inc, 2026) support AI-created agent topologies, which means domain experts have to manually design the agent structure. Further, the static agent structure lacks flexibility as a parent agent can only assign tasks to pre-defined sub-agents, and the sub-agents' informative states are discarded after execution. In contrast, OpenSage provides a comprehensive sub-agent management mechanism so that a parent agent in OpenSage can create sub-agent instances at runtime, allowing vertical and horizontal agent topologies.

**Tool.** While Claude ADK claims to support AI-written tools, its implementation is not open-sourced. Empirical performance suggests limited dependency awareness, which restricts the types of tools that can be constructed. It also lacks a layered organization, making it difficult to scale the system to support a large number of tools. For tool management, only OpenHands SDK and Claude ADK support native sandbox environments. However, they assume a single, shared execution environment, ignoring tools with conflicting dependencies or heterogeneous runtime requirements. In contrast, OpenSage supports AI-written tools and comprehensive management mechanisms.

**Memory.** None of the existing ADKs supports AI-created memory (Table 1). Our OpenSage design, however, lets AI itself decide what to persist. Instead of organizing memory into linear lists, OpenSage stores both short-term and long-term memory as queryable graphs with explicit relationships, so that AI can retrieve more complete and relevant knowledge than relying solely on embedding similarity. OpenSage features a dedicated memory agent to handle storing, retrieving, and updating memory.

## 3. Key Techniques

### 3.1. Overview

**Problem scope and key insights.** As discussed in Section 2, existing ADKs follow a common paradigm in which agent developers manually construct the entire agent structure and craft the tools and memories for each sub-agent. This paradigm requires substantial human effort and domain expertise, while imposing fundamental limitations on scalability and generalizability. This approach is analogous to early ML models that relied on feature engineering and hand-crafted model structures, which are ineffective compared to modern neural network models that learn representations directly from raw data with minimal human intervention.

In this paper, we propose OpenSage, the *first Agent Development Kit (ADK) that supports AI-centered agent construction, including automatically creating agent topology, designing tools, and managing memory*. Our core idea is to *provide a minimal yet essential scaffold that enables AI*

*to autonomously explore and construct agents*. Even if current models have not yet reached the level of intelligence required for such tasks, OpenSage serves as a scaffold and provides a training environment for future, more capable AI systems. This idea is also aligned with the fundamental trend in AI evolution: the shift from human-centered design toward AI-centered development, granting AI freedom to explore and learn global optimal solutions. Figure 2 provides an overview of OpenSage, which centers around the three critical components of an ADK.

**Self-generating agent topology.** Enabling AI to create agents and manage agent lifecycles are two major technical challenges. For creation, we propose a two-step procedure, where we first let the parent agent create an agent configuration that specifies the metadata of the sub-agents it intends to create. Then, we parse the agent configuration file and create a sub-agent as a Python object and store it in a unified sub-agent pool. This design balances efficiency and flexibility while simplifying the task for the parent agent. The agents are managed in the pool, including searching for existing sub-agents, registering new ones, and invoking stored sub-agents. We also design a graph-based memory for tracking and maintaining agent states. During runtime, sub-agents can communicate with each other, and their results can be integrated through the *agent ensemble* mechanism.

**Dynamic tool synthesis.** First, dynamically creating and registering tools is a new functionality that no existing ADK supports. In OpenSage, we register a set of existing meta-tools that enable agents to write new tools. The newly written tools are then registered through the existing meta-tools, which form a hierarchical tool structure and thus improve tool discovery efficiency for many tools. For example, we provide a Bash interface that allows agents to write new Bash commands as reusable tools. Second, managing dynamically generated tools is challenging, particularly because many tools are stateful and may take a long time to run. In OpenSage, we introduce tool-specific sandboxing to avoid conflicts between tool executions and support tool state management, enabling tool state saving and reusing. To improve efficiency, we support asynchronous tool execution, i.e., tools with long execution times run in the background, while agents periodically query tool states to monitor execution progress. Finally, we provide a domain-specific tool set tailored to software engineering tasks, including both static and dynamic program analysis tools.

**Hierarchical memory.** First, we support target-level long-term memory and execution-based short-term memory, where the long-term memory is a graph database that captures shareable, high-level knowledge across tasks on the same target. Our short-term memory is also a graph structure, which represents the spatial and temporal relationships of different agents' memories. This design simplifies

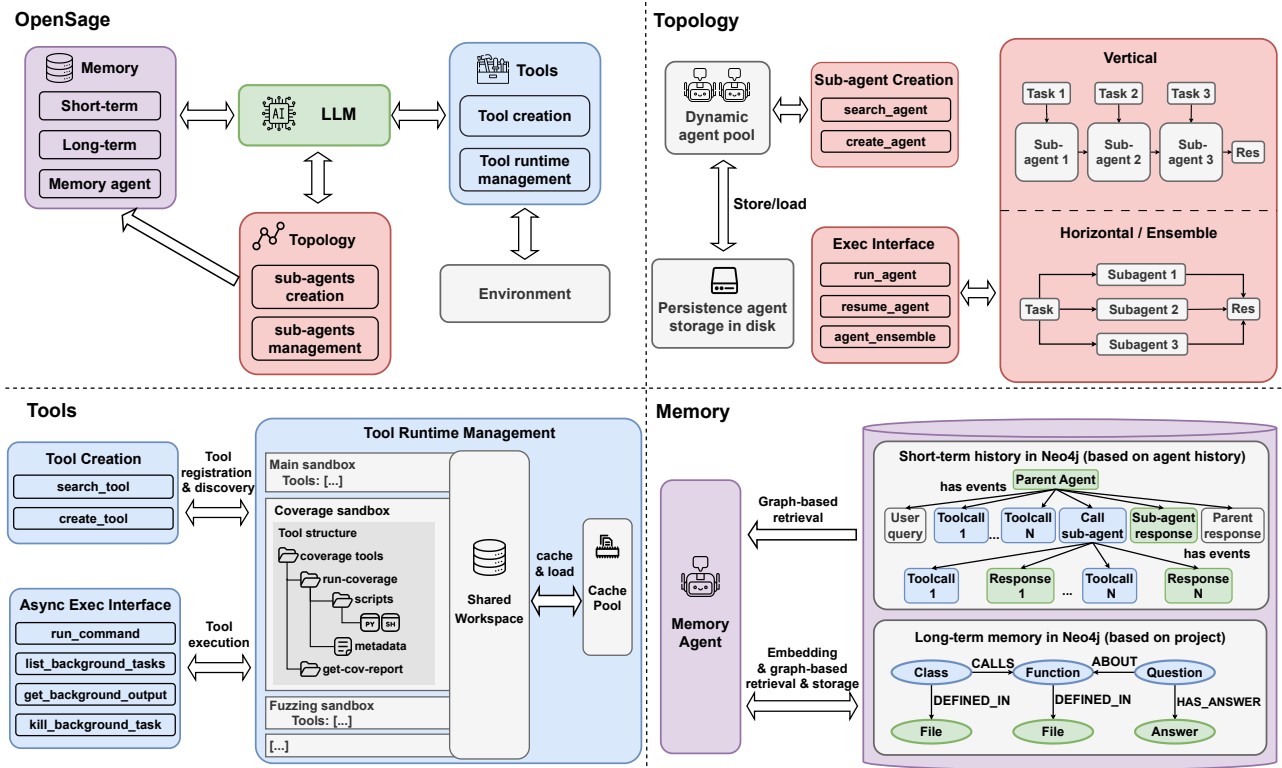

*Figure 2.* Overview of OpenSage framework, consisting of three key components. First, we enable AI to create different topologies while managing them in a unified agent pool. We then propose a hierarchical tool structure, including tool-specific sandboxing and states, and asynchronous execution. We design graph-based short-term and long-term memory with a memory agent to interact with them.

the state management for dynamically created agents, as each new agent is assigned an isolated memory instance by adding a new node to the graph. Second, following our AI-centered design principle, we introduce a general memory-agent abstraction. This agent is equipped with memory read and write tools that enable flexible memory management. Third, for memory retrieval, we support both graph-based and similarity-based mechanisms: the former enables coarse-grained localization within the graph, while the latter allows for fine-grained retrieval of specific items.

## 3.2. Self-generating Agent Topology

**AI-created agent topology.** We design agent creation as a tool that the parent agent can use to create sub-agents during run-time. The input is the metadata specifying the model name, system instruction, tools, a description, and initial memory state, the typical components of an agent (Durante et al., 2024). It then parses this metadata and constructs a sub-agent as a Python object and stores it in a unified sub-agent pool, which is used to store and invoke sub-agents. During agent creation, the tool configuration in the metadata is defined by either name or path. The tool set of the sub-agent is then initialized based on the specification, which is either inherited from the parent agent or retrieved from

the sandbox environment. As detailed in Section 3.4, each sub-agent maintains its own short-term memory, where the initial state can be empty or a summary of the parent agent. The sub-agent also has access to the long-term memory.

Each sub-agent can access all OpenSage's features, including creating new sub-agents. This yields a diverse space of agent topologies. As shown in Figure 2, two topologies are particularly useful. In the vertical topology, a parent agent delegates different sub-tasks to different sub-agents. This strategy helps isolate contexts to mitigate context overflow and restricts the set of available tools for each agent, preventing it from being overwhelmed by excessive tool choices. In the horizontal topology, multiple sub-agents work on the same task using distinct plans and later merge their results.

**Agent management.** OpenSage maintains all dynamically created sub-agents in a unified sub-agent pool (shown in Figure 2). We provide specific tools to list and search sub-agents in this pool by name or description, so that a parent agent can first attempt to reuse existing sub-agents before creating new ones. We also provide a tool for a parent agent to run a sub-agent, where the input specifies the sub-agent to run and the task to be executed, and the output is the corresponding sub-agent's response. During the process, the tool

finds the sub-agent's Python object, clones it, and executes the cloned object on the given task. The sub-agent's state is managed by the memory component described in Section 3.4, which supports resuming execution from a specific point in time. When the sub-agent is finished, the corresponding cloned Python object is deleted.

To avoid race conditions in horizontal topology, OpenSage enables agent communications, where parallel sub-agents are aware of each other through generated prompts and share a message board file. Sub-agents can write to the board with locks. OpenSage monitors the board, tracks the portion each sub-agent has read, and piggybacks message diffs onto tool responses to keep sub-agents synchronized.

**Agent ensemble.** We implement this mechanism as a tool whose input includes the task description, the selected list of sub-agents, and the model assigned to each sub-agent. When a parent agent calls this tool, OpenSage locates the specified sub-agents in the sub-agent pool, clones their Python objects, updates them with the designated models, and runs them in parallel. Once all sub-agents complete their execution, their responses are summarized and returned to the parent agent.

### 3.3. Dynamic Tool Synthesis

**Tool creation and organization.** OpenSage supports the dynamic creation and registration of tools. As the tool set grows, effective organization and discovery become essential. To this end, OpenSage adopts a file-system-based hierarchical structure (Figure 2) that scales naturally for tool discovery and registration. Each tool is represented as a module (e.g., a Python module or a Bash script) stored in the file system, accompanied by metadata that specifies its description, interface, and dependencies. Each directory level also includes documentation that summarizes the tools and sub-tool sets it contains. At runtime, the agent first narrows down relevant tool categories and then inspects candidate tools, reducing context load and enabling keyword-based search. During agent initialization, only top-level tool-module information is loaded, avoiding the context overhead of enumerating tens or hundreds of tools. Dynamic tool creation is enabled through a set of meta-tools that provide primitives for tool synthesis. When an agent creates a new tool, it generates both the implementation and the corresponding metadata. The OpenSage runtime validates the metadata and registers the tool into the active tool set. This programmable interface allows agents not only to invoke tools but also to inspect, modify, and extend existing tools based on task requirements, providing substantially greater flexibility than static tool APIs.

**Runtime tool management.** OpenSage provides container-based execution and state management to support tools with heterogeneous compilation and runtime requirements (Figure 2). Each tool set specifies its environment requirements (e.g., programming language runtimes and system dependencies) in metadata, and OpenSage automatically provisions an isolated Docker container with the appropriate configuration. It allows tools with conflicting dependencies to coexist and prevents interference with the agent's execution environment. OpenSage mounts a shared workspace via Docker volumes (Docker, 2026) to support data sharing across containers. To reduce setup overhead, OpenSage commits container snapshots as Docker image layers after initialization or execution, capturing installed packages, compiled artifacts, and intermediate files. Subsequent invocations reuse these cached states, substantially cutting startup time for tools with expensive setup (e.g., building large codebases or initializing analysis frameworks). OpenSage also provides an asynchronous execution interface for long-running tools without blocking agent reasoning. If an invocation reaches its time limit or is explicitly designated as a background task, it is offloaded to run asynchronously. Background invocations return a handle, analogous to a process ID, which can be used to poll status, retrieve results, or terminate runs. This design is important for compute-intensive tools such as static analysis and compilation, which may run for extended periods with minimal intervention.

Finally, OpenSage includes a domain-specific toolkit enabling both static and dynamic program analysis, which improves the agents' capabilities in coding and software engineering tasks. This toolkit is summarized in Table 3.

### 3.4. Hierarchical Memory Management

**Short-term memory.** As described in Section 3.2, our dynamic sub-agent creation naturally produces a hierarchical execution structure, which motivates the design of a graph-based short-term memory. As shown in Figure 2, the graph consists of nodes and edges starting from a parent agent (represented by an `AgentRun` node). This parent agent creates step-level tool calls and responses, which are stored as `Event` nodes. Every time a sub-agent is created, a new `AgentRun` node is opened, which then generates its own event nodes. Long tool outputs are summarized, with full outputs stored as `RawToolResponse` nodes referenced from summarized events. Furthermore, older history can be compressed into summary events when the context grows too large. Edges connecting the agents represent tool calls as well as cross-agent calls. On top of this graph, OpenSage provides tools for retrieval, which can list sub-agent executions, inspect events, recover unsummarized outputs, and perform low-level Neo4j-based graph queries. Such tools will be queried by our memory agents during execution.

**Long-term memory** is designed to capture higher-level knowledge about the targets that can be shared across different tasks. As shown in Figure 2, OpenSage represents long-term memory as a graph managed by Neo4j and stored

in a separate database. Each node corresponds to an entity (e.g., code structures such as classes and functions, user queries and answers, or other relevant concepts) and directed edges represent the relationships between these entities. This graph is iteratively constructed as the memory agent invokes a set of storage tools, including creating nodes and edges and listing existing node and edge types. When creating a node, the tool takes as input a node type, a label, and the content. The label denotes a keyword or question, and the content represents the corresponding description or answer. During this process, the tool computes an embedding of the label using text-embedding-3-small and creates a node of the specified type with the label, content, and embedding stored in the graph. We define a set of node and edge types tailored to coding-oriented tasks, while for non-code scenarios, the model can propose appropriate types on its own. To create an edge, the tool takes a source node, a target node, and an edge type, and then inserts a directed edge of the specified type between the two nodes to record their relationship. For retrieval, OpenSage provides two kinds of tools. The first takes as input a target node type and a query label, embeds the label, and returns the top-$N$ matching nodes of that type together with their one-hop subgraphs. The second performs pattern-based lookup over node labels using grep-style matching.

**Memory agent.** The memory agent serves as a bridge between user-built agents and the underlying memory graphs. A user-built agent does not need to understand the internal schema; instead, it issues natural language instructions, and the memory agent performs the appropriate operations. Upon receiving a query, the memory agent first decides whether it targets short-term or long-term memory. For short-term memory, which only supports search, it iteratively invokes the retrieval tools to get the requested execution history. We do not enable the memory agent to write to short-term memory because 1) short-term memory is automatically updated during execution, 2) allowing the memory agent to modify it could break the context structures. For long-term memory, the memory agent supports search, update, and store operations. For search-related queries, it extracts key entities from the request, invokes the retrieval tools over the long-term graph, and aggregates the results into a concise summary. For update and store queries, it first searches existing nodes, then decides whether to add, modify, or delete nodes and edges, so that only non-redundant, relevant knowledge is persisted.

## 4. Evaluation

### 4.1. OpenSage on SOTA Benchmarks

**Benchmarks.** To evaluate self-generating agent topology and our tooling system, we select CyberGym (Wang et al., 2025c). This large-scale benchmark features 1,507 real-

world C/C++ vulnerabilities, where the agent must craft proof-of-concept (PoC) input for vulnerability reproduction. CyberGym presents complex reasoning tasks that naturally decompose into sub-tasks, require extensive domain-specific knowledge about security vulnerabilities, and demand specialized tools and containerized environments for execution.

To test whether OpenSage generalizes across heterogeneous task domains, we select Terminal-Bench 2.0 (The Terminal-Bench Team, 2025). This benchmark comprises 89 expert-curated terminal tasks in containerized environments, spanning diverse categories (e.g., SWE, scientific computing, ML) of high-skill tasks. We run experiments with 5 trials using the official evaluation framework (Shaw, 2025), following all specified time and compute constraints. We further benchmark on DevOps-Gym (Tang et al., 2026), the first benchmark covering the complete DevOps cycle with 705 real-world tasks across build and configuration, monitoring, issue resolving, and test generation, plus 17 end-to-end pipeline tasks that chain all four stages sequentially.

We select SWE-Bench Pro (Deng et al., 2025) to evaluate our memory mechanism on long-horizon tasks, which require agents to maintain and retrieve context over extended trajectories. Python is the only programming language that is supported by all major SWE agents (including our baseline), and it is the most widely used language. Hence, we run our experiment on all 266 Python tasks in SWE-Bench Pro. Moreover, we further evaluate OpenSage's hierarchical memory management on long-term dialogues using LO-COMO in Appendix D, thereby demonstrating the generality of our memory design.

**OpenSage agent design.** We select the backbone model based on leaderboard results, prioritizing models that perform well and can effectively leverage our proposed features. We also consider the balance between throughput, cost, and performance, as both CyberGym and SWE-Bench Pro are large-scale benchmarks. For CyberGym, we enable the tooling system along with the domain-specific tool set and the self-generating agent structure. For Terminal-Bench 2.0, we develop an agent without the self-generating agent structure, since (i) many tasks are straightforward and do not require multiple stages, and (ii) strict resource constraints (e.g., CPU limits during compute-intensive operations such as password brute-forcing) limit the benefits of parallel exploration; under such constraints, parallel exploration may even incur additional overhead. For SWE-Bench Pro, we design a coding agent with our hierarchical memory features enabled and prompt it to first launch a sub-agent that explores the codebase and populates the long-term memory before solving the issue, and to read from and write to memory during issue resolution.

**Baselines.** For each benchmark, we compare against top-performing agents (e.g., Ante (Antigma Labs, 2026), SWE-

*Table 2.* Comparison of overall performance of agents built with OpenSage against other state-of-the-art agents and ADKs. Open-Sage agents (denoted as SageAgent) **rank first** on the leaderboard for CyberGym, DevOps-Gym and Terminal-Bench 2.0.

| Benchmark | Agent | Model | ADK | % Resolved |
|-----------|-------|-------|-----|-----------|
| CyberGym | SageAgent | GPT-5 (medium) | OpenSage | 60.2 |
| | Anthropic Agent | Claude Opus 4.5 | Claude | 50.6 |
| | OpenHands | GPT-5 (high) | OpenHands | 39.4 |
| | Anthropic Agent | Claude Sonnet 4.5 | Claude | 28.9 |
| Terminal-Bench 2.0 | SageAgent | GPT-5.3-Codex | OpenSage | **78.4 ± 2.2** |
| | Droid | GPT-5.3-Codex | - | 77.3 ± 2.2 |
| | Simple Codex | GPT-5.3-Codex | OpenAI | 75.1 ± 2.4 |
| | Claude Code | Claude Opus 4.6 | Claude | 58.0 ± 2.9 |
| | OpenHands | Claude Opus 4.5 | OpenHands | 51.9 ± 2.9 |
| SWE-Bench Pro | SageAgent | Gemini 3 Flash | OpenSage | **59.0** |
| | SWE-agent | Gemini 3 Flash | - | 40.2 |
| | Agentless | Gemini 3 Flash | - | 9.4 |
| DevOps-Gym | SageAgent | GPT-5.3-Codex | OpenSage | **46.8** |
| | Claude Code | Claude Sonnet 4 | Claude | 27.5 |
| | OpenHands | Claude Sonnet 4 | OpenHands | 21.0 |

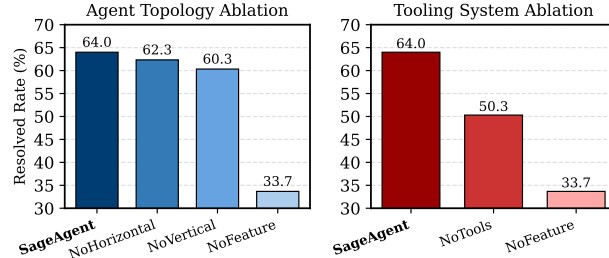

*Figure 3.* Ablation analysis of SageAgent built with OpenSage framework on a 300-instance subset of CyberGym, evaluating the impact of agent topology (left) and tooling system (right).

agent (Yang et al., 2024)) reported on the public leaderboard and representative agents built using popular ADKs, including Claude (Anthropic, 2026a), OpenAI (OpenAI, 2021), OpenHands (Wang et al., 2025b), etc.

**Results.** As shown in Table 2, SageAgent consistently outperforms the baselines. Notably, on CyberGym, DevOps-Gym and Terminal-Bench 2.0, SageAgent *ranks first* on the leaderboard. Furthermore, SageAgent is the only agent capable of solving end-to-end pipeline tasks on DevOps-Gym, achieving a 17.7% resolved rate, while all other baselines score 0%. Compared to OpenHands on CyberGym, SageAgent achieves a resolved rate that is *over 20%* higher, even when OpenHands uses the same backbone model with a higher reasoning effort setting. This improvement stems from our tooling system with domain-specific toolkits and the self-generating agent structure; we further analyze the effectiveness of each component in Sections 4.2 and 4.3. Compared with Droid and Simple Codex on Terminal-Bench 2.0, SageAgent achieves superior performance using the same backbone model, demonstrating that OpenSage delivers strong results even with only basic features enabled. On SWE-Bench Pro, SageAgent also outperforms the SWE-agent baseline, demonstrating the effectiveness of our hierarchical, agentic memory design for long-horizon software engineering tasks.

### 4.2. Evaluation of Self-Generating Agent Structure

**Objective.** We aim to evaluate the effectiveness of the self-generating agent structure of OpenSage through two groups of ablation studies on horizontal and vertical agent topologies, conducted on a 300-instance subset of Cyber-Gym. CyberGym is well-suited for this evaluation because it comprises long-horizon vulnerability analysis tasks that naturally decompose into sub-tasks.

**Ablation variants.** We evaluate three ablation configurations: 1) *NoHorizontal*: disables the agent ensemble, i.e.,

no horizontal agent topology; 2) *NoVertical*: disables dynamic sub-agent creation, i.e., no vertical agent topology; 3) *NoFeature*: disables all OpenSage features, including the tooling system and the self-generating agent structure, serving as a lower-bound baseline.

**Results.** With all the features enabled, we observe the model actively creating sub-agents for different sub-tasks with tailored instructions and dedicated toolsets, e.g., a debugging sub-agent shown in Appendix C.1. As shown in Figure 3, OpenSage with agent ensemble achieves a higher resolved rate than NoHorizontal. On the 27 tasks where it is triggered, the ensemble resolves 15% more instances, indicating its effectiveness. Comparing OpenSage with NoVertical reveals that removing this capability leads to a substantial performance drop. Without dynamic sub-agent creation to decompose tasks and isolate context, the context length frequently exceeds the context window and triggering summarization that loses important details. The average number of summarization events per task increases from 6.4 to 13.1, indicating substantially greater information loss. Moreover, logically unrelated tool calls accumulate in the shared context, making it harder for the model to reason effectively. However, we also observe cases (Appendix C.3) where the agent creates sub-agents with mismatched toolsets and purposes, hallucinated tools and sub-agents, or overly complicated instructions, reducing the effectiveness.

**Comparison with expert-designed topology.** To demonstrate the effectiveness of self-generated agent structures, we further compare OpenSage with Agentless (Xia et al., 2025) on SWE-Bench Pro. Agentless employs an expert-designed workflow and serves as an essential baseline for software engineering tasks. Because Agentless only supports Python, we evaluate on the Python subset of 266 instances, where OpenSage achieves a resolved rate of 59.0%, far higher than Agentless (9.4%). Despite being implemented by 6.3K lines of Python code, Agentless is still fundamentally limited: its fixed workflow prevents the agent from retrieving information on demand, provides poor support for multi-file edits, and disallows patch refinement, which together lead to its

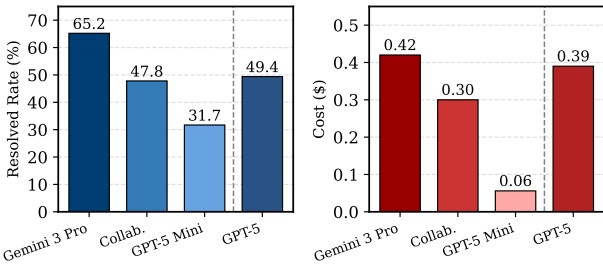

*Figure 4.* Resolved rate (left) and cost (right) for agents built with OpenSage framework on Terminal-Bench 2.0 using Gemini 3 Pro, GPT-5 Mini, and a large-small collaboration setup (Gemini 3 Pro + GPT-5 Mini), compared against GPT-5.

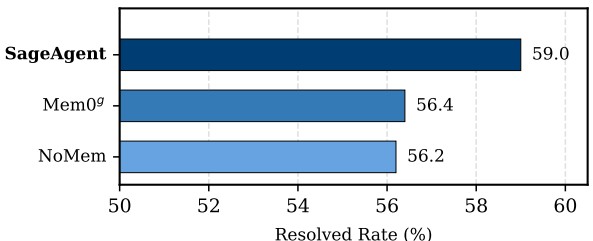

*Figure 5.* Ablation analysis of SageAgent built with OpenSage framework, comparing different memory designs (agentic, Mem0$^g$, no memory) on SWE-Bench Pro.

low performance. This fundamentally stems from human experts defining the agent's behavior through rule-based code, rather than letting AI decide when and how to act. In contrast, on top of OpenSage's generic framework, our SWE-Bench Pro agent requires only 531 additional lines of code, yet achieves much better results.

**Heterogeneous model collaboration.** To further demonstrate the flexibility of OpenSage, we evaluate a large-small collaboration pattern on Terminal-Bench 2.0 in which a strong model handles planning and autonomously creates sub-agents with a weaker model to perform detailed implementation. We choose Terminal-Bench 2.0 for this evaluation due to its large task diversity. As shown in Figure 4, pairing Gemini 3 Pro (planning/review) with GPT-5 Mini (execution) substantially improves accuracy over GPT-5 Mini alone, matching GPT-5's performance while reducing cost compared to Gemini 3 Pro alone.

### 4.3. Evaluation of Tooling System

**Objective.** We aim to assess the contribution of OpenSage's tooling system, including the domain-specific toolkit and dynamic tool creation, through ablation studies conducted on the same 300-instance subset of CyberGym as in Section 4.2. The CyberGym benchmark is well-suited for this analysis because it requires agents to perform diverse security analysis tasks, demanding heterogeneous domain-specific tools and creating new tools at runtime. Evaluating the toolkit and dynamic tool creation separately requires non-trivial efforts, as we design the tooling system around the principle of self-programming. The agent benefits from developing new tools based on existing effective tools; disabling OpenSage's tool management and preventing dynamic tool creation would make the toolkit unusable in practice. This tight coupling between tool creation, management, and execution also explains why existing ADKs cannot support such a heterogeneous, dynamically managed toolset.

**Ablation variants.** We evaluate two variants: 1) *NoTools*: replaces the entire tooling system with a raw terminal in-

terface; 2) *NoFeature*: disables all features, including the tooling system and the self-generating agent structure.

**Results.** As shown on the right of Figure 3, we observe a substantial performance drop from OpenSage to No-Tools, confirming the effectiveness of OpenSage's tooling system. Instead of relying solely on initially provided general-purpose tools, the agent creates tools tailored to specific scenarios. On this 300-instance subset, OpenSage creates 39 tools written in Python and C/C++, including grammar-aware fuzzers, seed generation and mutation utilities, and file-format-specific input generators, demonstrating that OpenSage's dynamic tool synthesis and management mechanisms are effectively exercised in practice, rather than relying on the general-purpose tools provided. Moreover, the additional degradation from NoTools to NoFeature consistently highlights the importance of OpenSage's self-generating agent topology.

### 4.4. Evaluation of Memory Management

**Objective.** We aim to assess the effectiveness of Open-Sage's memory mechanisms through ablation studies on SWE-Bench Pro. SWE-Bench Pro is well-suited for this evaluation because it comprises long-horizon SWE tasks that require agents to maintain and retrieve relevant context across extended execution trajectories.

**Ablation variants.** We evaluate two alternative configurations: 1) *NoMem*: disables memory management entirely; 2) *Mem0$^g$*: equips the agent with the SOTA Mem0$^g$ memory (Chhikara et al., 2025), which leverages a graph-based memory structure for context storage and retrieval without using any AI-centered memory management.

**Results.** As shown in Figure 5, OpenSage with our hierarchical memory achieves the best resolved rate on SWE-Bench Pro, substantially outperforming the OpenSage variant without our memory design, demonstrating the effectiveness of our hierarchical memory, featured by AI-created memory and AI-driven memory management. Mem0$^g$, however, brings little improvement over NoMem: its node re-

lationships are hard-coded and cannot adapt to tasks like SWE-Bench Pro, and it fully relies on the model to invent node types without any way to list or constrain them, which leads to ungrounded note types and prevents its memory from organizing complex, structured information. In contrast, OpenSage truly leverages AI-created memory and AI-driven memory management by providing a set of node and edge types tailored specifically for coding tasks, enabling more structured and semantically meaningful knowledge, while still maintaining generalizability by allowing the agent to create new node/edge types but offering mechanisms to list them, as demonstrated in Appendix D. Additionally, OpenSage supports pattern-based lookup over node labels, allowing exact symbol matching that complements embedding-based retrieval. Concrete examples can be found in Appendix C.2.

## 5. Conclusion and Future Work

In this paper, we present OpenSage, the first agent development kit that enables AI to autonomously construct agent topologies based on given tasks with flexible toolsets and comprehensive memory support. We evaluate OpenSage across four SOTA benchmarks, demonstrating its superiority over existing ADKs and validating the importance of its core system designs.

This work highlights several promising directions for future research. First, we plan to extend OpenSage to support AI-generated workflows. For example, we will provide functions enabling AI to construct parallel workflows by allowing LLMs to determine dependencies and communication protocols across agents. Second, we plan to incorporate model training support on top of OpenSage. On the one hand, we will provide convenient rollout interfaces for post-training frameworks such as AReaL (Fu et al., 2025), verl (Sheng et al., 2024), and LlamaFactory (Zheng et al., 2024). On the other hand, we will support a Kubernetes-based (Kubernetes, 2019) sandbox backend to run many environments in parallel, enabling large-scale data collection and training on real-world tasks.

## Acknowledgements

We gratefully acknowledge support from UC Noyce Initiative, Coefficient Giving, FAR AI, OpenAI, Anthropic, Google, and Berkeley RDI.

## Impact Statement

OpenSage advances the design of agent development kits by moving from manually engineered, fixed agent structures toward AI-centered construction of agents, tools, and memory. We expect the primary societal impact to come from lowering the engineering barrier for building robust, tool-augmented agents, enabling researchers and practitioners to more easily prototype, evaluate, and iterate on complex agentic systems. By integrating self-generated agent topology, dynamic-created tooling, hierarchical memory, and containerized execution into a unified framework, Open-Sage can help standardize infrastructure that is currently rebuilt in an ad hoc manner across projects, potentially improving reliability and reproducibility. At the same time, we emphasize the importance of adopting appropriate safeguards, auditing practices, and organizational policies when deploying such systems in real-world environments, so that the increased accessibility of powerful agents is aligned with responsible and transparent use.

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

*Table 3.* Domain-specific toolkit for software engineering and security tasks.

| Category | Tool set | Libraries | Features |
|---|---|---|---|
| Static | Code analysis | Joern (joern, 2024), CodeQL (GitHub, 2026b) | Code property graph query, call graph analysis, dataflow-based program slicing, semantic-aware code search |
| Dynamic | Fuzzing | AFL++ (Heuse et al., 2022), LibFuzzer (LLVM, 2026a) | Customizable seed generation, mutation, and scoring |
| | Coverage | LLVM-Cov (LLVM, 2026b) | Query test case coverage with Neo4j, generate detailed reports |
| | Debugger | GDB (Free Software Foundation, 2026), PDB (Python, 2026) | Set breakpoints, inspect program states, trace program execution, custom debugger commands |

# A. More Related Work

**Coding and software engineering agents.** As a promising application domain, numerous agents have been developed for coding tasks and software engineering tasks (Wang et al., 2024; Anthropic, 2026a; Anysphere, 2026; Google, 2026; GitHub, 2026a). These agents resolve issues (Yang et al., 2024; Zhang et al., 2024; Bouzenia et al., 2024; Li et al., 2025; Trae Team, 2026; Yang et al., 2025; JD AI Research, 2026), detect and patch vulnerabilities (Potter et al., 2025; Kim et al., 2025; Sheng et al., 2025; Luo et al., 2026; Dai & Xie, 2025; Tang et al., 2025; Guo et al., 2025; Li et al., 2024; Nie et al., 2025), and perform penetration testing and CTF competition (Deng et al., 2024; NYU-LLM-CTF Team, 2026; Abramovich et al., 2025). Despite being designed by world-class experts, many of these agents suffer from fundamental limitations, such as rigid, pre-defined agent structures and static toolsets. Notably, most coding agents do not even support debuggers. These shortcomings stem from foundational constraints in current ADKs. As we will show in Section 4, addressing these limitations in our OpenSage enables the construction of agents that significantly outperform existing ones.

# B. Details of Domain-Specific Toolkit

Table 3 shows the domain-specific toolkit for software engineering and security tasks, including static and dynamic analysis tools.

# C. Examples of Agent Trajectories

## C.1. Self-generating Agent Topology and Tooling System

We study the case of `arvo:14574` in CyberGym with GPT-5 as a simple example to illustrate the behavior of OpenSage's self-generating agent topology and tooling system.

### C.1.1. TASK BACKGROUND

In this case, the vulnerability is in libarchive's RAR5 decompression code and is triggered through the following crashing call chain: `process_block → parse_tables → decode_number → read_bits_16`. At a high level, libarchive reads the archive as a sequence of blocks. Some blocks may carry a small decoding table (a Huffman table) that is required before the block's compressed bytes can be interpreted. The crash happens when an input block claims that a Huffman table is present, but the block does not actually contain enough bytes for that table. Libarchive then follows this chain of functions: 1) `process_block`: sees the "table present" flag and decides it must load the table. 2) `parse_tables`: attempts to read the table data from the beginning of the block. 3) `decode_number`: repeatedly decodes values while building the table. 4) `read_bits_16`: reads several bytes from the current block buffer to extract the next bits. The core problem is that `read_bits_16` assumes the block buffer contains enough bytes and directly reads three bytes from it. When the "table present" flag is set, but the table bytes are missing (effectively a zero-length or too-short table region), these reads go past the end of the buffer, causing an out-of-bounds memory read.

C.1.2. AGENT BEHAVIOR

The agent's trajectory can be divided into the following stages.

**Tool discovery and initial code exploration.** At the beginning of the task, the agent loads detailed descriptions of static analysis tools, and then uses the static analysis tools as well as general bash tools (grep, sed) to inspect files and code relative to the issue description. In this stage, the agent sees all functions in the crashing call chain and realizes the vulnerability.

**Initial PoC generation and misunderstanding** The agent only noticed that constructing an abnormal Huffman table and reaching the decompression logic in parse_tables would trigger the vulnerability, but it did not analyze the branch conditions required to reach parse_tables. It generates a PoC, but configures the compressed block without enabling a "Huffman table" flag. This causes the program not to follow the crashing call chain, but instead to enter a branch in process_block that does not invoke parse_tables, as shown in Listing 1.

```
1  static int process_block(struct archive_read* a) {
2      ...
3      if (/* the current block's header contains a Huffman table flag */) {
4          ret = parse_tables(a, rar, current_block_data); /* Load Huffman tables. */
5          ...
6      }
7      ...
8  }
```

*Listing 1.* Code snippet in process_block

**Sub-agent creation and debugging.** Since the generated PoC did not trigger a crash, the agent plans to debug the program and verify whether the actual execution path matches its expectation. However, since debugging is a separate sub-task, the agent decides to first create a dedicated agent named *gdb_helper* for this debugging task and provide it with debugging-related tools, and then calls the sub-agent to check the execution path Listings 2 to 4. The sub-agent calls tools including set_file, set_input_file, set_breakpoint, step_control, execute, and other debugging-related tools to fulfill the task. Its responses enable the parent agent to find the location where the actual execution starts to deviate from its expectation.

```
{
  "function_call":
  {
      "name": "create_agent",
      "args": {
          "agent_name": "gdb_helper",
          "role": "GDB debugging assistant",
          "instruction": "Load binaries, set breakpoints, run with the provided PoC, and
              trace execution from the fuzzer entrypoint to the RAR5 parser.",
          "tools": ["set_file", "step_control", "set_breakpoint", ...],
      }
  }
}
```

*Listing 2.* Tool call for creating a sub-agent for debugging

```
{
  "function_call":
  {
      "name": "call_agent",
      "args": {
          "agent_name": "gdb_helper",
          "task_message": "Load /out/libarchive_fuzzer in GDB, set args to /shared/tmp/
              agent/poc, set breakpoints at uncompress_file, do_uncompress_file,
              process_block, parse_tables, decode_number, read_bits_16. Then run the
              program until the first breakpoint is hit, continue execution until the
              program finishes, and report which breakpoints were hit."
      }
  }
}
```

*Listing 3.* Tool call for invoking the debugging sub-agent

```
{
  "function_call":
  {
    "name": "call_agent",
    "response": {
        "agent": "gdb_helper",
        "status": "success",
        "summary": [
          "Program finished. The RAR5 parser was successfully entered from the fuzzer
              entrypoint...",
          "..."
        ],
        "observations": [
          "breakpoint_hit: uncompress_file, do_uncompress_file, process_block",
          "symbol_status: debug symbols stripped..."
        ],
    }
}
```

*Listing 4.* Initial debugging response from the GDB sub-agent

**Successful PoC.** The agent then inspects the code of `process_block`, corrects the previously incorrect branch condition in the PoC, and generates a working PoC.

## C.2. Memory Management

We use a case in SWE-Bench Pro with Gemini 3 Flash to illustrate how OpenSage's hierarchical memory management works, including how it stores and retrieves memories.

### C.2.1. TASK BACKGROUND

In this run, the agent is solving a software-maintenance task (an Ansible Galaxy validation bug). Concretely, the bug is that Ansible's collection-name validation can accept Python reserved keywords in the namespace or collection name (e.g., `def.collection`), which should be rejected to avoid ambiguous or unusable identifiers.

### C.2.2. AGENT BEHAVIOR

**Storing high-signal outcomes as structured memory.** At the beginning of each run, the agent first explores the codebase and then decides whether to persist its findings in memory for later reuse. During the whole issue-resolving task, the agent also automatically stores high-signal intermediate outcomes. In this run, the system stores multiple high-level findings that are useful beyond the immediate tool output: (i) distilled search results (e.g., Listing 5), (ii) general code understanding (e.g., Listing 6) and (iii) environment errors (e.g., Listing 7).

```
{
    "function_call":
    {
      "name": "save_memory",
      "args": {
          "node_type": "search_result",
          "label": "Search for 'validate|keyword'",
          "content": "Search results for 'validate|keyword' in Ansible Galaxy code,
              highlighting locations related to certificate validation and Python keyword
              checks within role and collection management
          ...
          "
      }
    }
}
```

*Listing 5.* Storing a summarized search result into memory

```
{
    "function_call":
    {
      "name": "save_memory",
      "args": {
          "node_type": "code_understanding",
          "label": "Logic for Ansible collection validation and galaxy.yml checks",
          "content": "### Relevant Code
          '/app/lib/ansible/galaxy/collection.py': Contains the 'validate_collection_name'
              function and 'build_collection' function.
          '/app/lib/ansible/utils/collection_loader/_collection_finder.py': Defines
              validation regexes and contains 'AnsibleCollectionRef'.
          '/app/lib/ansible/galaxy/dependency_resolution/dataclasses.py': Includes '
              _is_fqcn' and references 'galaxy.yml'.
          '/app/lib/ansible/cli/galaxy.py': CLI entry points (e.g., 'galaxy init') calling
               'validate_collection_name'.
          '/app/lib/ansible/galaxy/data/collections_galaxy_meta.yml': Schema-level
              constraints for 'galaxy.yml' fields such as 'namespace'.
          ### How the Validation Works..."
      }
    }
}
```

*Listing 6.* Storing collection validation-related logic into memory

An environment error is categorized and stored as an `error` memory item:

```
{
    "function_call":
    {
      "name": "save_memory",
      "args": {
          "node_type": "error",
          "label": "Missing PyYAML dependency for Ansible collection validation",
          "content": "Traceback indicates a ModuleNotFoundError: No module named 'yaml'
              when trying to import 'validate_collection_name' from 'ansible.galaxy.
              collection'. This suggests a missing dependency for Ansible's collection
              validation functionality."
      }
    }
}
```

*Listing 7.* Storing an environment error into memory

**Retrieval at the point of need.** Later in the same run, the context window is exceeded, the agent's history is summarized, and the agent queries its memory store using search_memory.

```
{
    "function_call":
    {
        "name":"search_memory",
        "args": {
            "query":"search the collection name validation mechanism"
        }
    }
}
{
    "function_response":
    {
        "name":"search_memory",
        "response":{
            "found": true,
            "total_found": 5,
            "results": [
```

```
              {"node_type": "code_understanding", "label": "Logic for Ansible collection
                  validation and galaxy.yml checks", "content": "### Relevant Code
                  ..."},
              {"node_type": "search_result", "label": "Search for 'validate|keyword'", "
                  content": "Search results ..."},
              {"node_type": "file", "label":"/app/lib/ansible/galaxy/data/
                  collections_galaxy_meta.yml", "content": "This file defines the schema
                   for Ansible Galaxy collection metadata (galaxy.yml) ..."}
              ...
          ]
      }
}
```

*Listing 8.* Example: the agent retrieves previously exploration summaries and related information.

### C.3. Misuse of OpenSage's Features

The following examples show common patterns in the misuse of OpenSage's features by the SOTA model GPT-5 on CyberGym.

### C.3.1. HALLUCINATIONS ABOUT TOOLS

In the example shown in Listing 9, the model hallucinates a tool named applypatch that does not exist. This is later resolved as the model calls the cat to rewrite the whole file.

```
{
    "function_call":
    {
        "name": "run_terminal_command",
        "args": {
          "command": "applypatch << 'PATCH'\n\
            *** Begin Patch\n\
            *** Add File: /shared/tmp/agent/gen_tmap.c\n\
            // Generator for a valid XCDR2 DDS_XTypes_TypeMapping PoC\n\
            #include <stdint.h>\n\
            ...
            *** End Patch\n\
            PATCH"
        }
    }
}
```

*Listing 9.* Hallucinated tool invocation of applypatch which does not exist.

### C.3.2. HALLUCINATIONS ABOUT SUB-AGENTS

In the example shown in Listing 10, the model first hallucinates a sub-agent named generic_consultant and attempts to invoke it even though no such agent exists, resulting in a failed call. This misuse is later resolved as the model correctly creates a dynamic sub-agent named generic_consultant and then invokes it through the proper interface.

### C.3.3. MISALIGNED SUB-AGENT TOOL SCOPES

As shown in Listing 11, the sub-agent is explicitly created to perform static analysis of libecc error paths, yet it is provisioned with an overly permissive tool scope that includes dynamic analysis tools such as gdb, coverage tracking, and fuzzing. In practice, no negative effects are observed, as the sub-agent's instruction and subsequent prompts constrain it to purely static reasoning.

## D. Evaluation on LOCOMO

**Setup and design.** The LOCOMO (Maharana et al., 2024) dataset is designed to assess long-term conversational memory in dialogue systems. It consists of 10 long conversations, each spanning multiple sessions with roughly 600 dialogue turns

```
# (1) Tool call: query available agents for ensemble
{
  "function_call":
  {
    "name": "list_active_agents",
    "args": {}
  }
}

# (2) Tool response: no available agents found
{
  "function_response":
  {
    "name": "list_active_agents",
    "response":{
        "summary": {
        "total_active_agents": 0,
        },
       "message": "Found 0 total agents. If no suitable agents exist, create a dynamic sub
           -agent."
    }
  }
}

# (3) Attempted invocation of a hallucinated sub-agent
{
  "function_call":
  {
    "name": "call_agent",
    "args": {
        "agent_name": "generic_consultant",
        "instructions": [
          "Analyze lwan-request.c::parse_headers() for potential out-of-bounds access.",
          "Derive precise byte-level preconditions leading to reading past buffer_end."
        ],
        "history_passed_in": false
    }
}

# (4) Failure response: agent not found
{
  "function_response":
  {
    "name": "call_agent",
    "response":{
      "status": failed,
      "error": "Agent 'generic_consultant' not found.",
      "summary": "No suitable agents available. Create a dynamic sub-agent and invoke it
          via the agent ensemble."
    }
  }
}
```

*Listing 10.* Hallucinated sub-agent invocation when no agents are available

```
{
    "function_call":
    {
      "name": "create_agent",
      "args": {
          "agent_name": "libecc_static_reader",
          "instruction": "You are a precise static-analysis assistant focused on C/C++
              lifetimes in libecc and cryptofuzz modules. Your job: (1) enumerate
              functions in libecc where local fp variables are declared; (2) identify
              error paths where fp_uninit is called on a local fp that may not have been
              initialized and whose .magic was not explicitly set to 0 before early
              returns; (3) extract exact control-flow conditions and inputs that trigger a
               goto err before initialization; (4) map reachability from cryptofuzz/
              modules/libecc/module.cpp operations, especially ECC_ValidatePubkey and
              LoadPoint; (5) derive concrete preconditions on fuzz input bytes to hit the
              path. Do not make assumptions; cite exact lines and conditions.",
          "model_name": "inherit",
          "tools_list": [
            "gdb_mcp",
            "retrieval",
            "static_analysis",
            "coverage",
            "fuzz"
          ],
          "description": "Sub-agent specialized in libecc static code-path analysis to
              detect uninitialized fp_uninit usage."
      }
    }
}
```

*Listing 11.* Creation of a static-analysis sub-agent with an inaccurately restricted tool scope

*Table 4.* Performance comparison of memory systems across different question types in the LOCOMO dataset.

| Method | Single Hop | Multi-Hop | Open Domain | Temporal |
|---|---|---|---|---|
| A-Mem* | $62.23 \pm 0.75$ | $47.92 \pm 0.47$ | $71.12 \pm 0.20$ | $23.43 \pm 0.39$ |
| Zep | $61.70 \pm 0.32$ | $41.35 \pm 0.48$ | $76.60 \pm 0.13$ | $49.31 \pm 0.50$ |
| OpenAI | $63.79 \pm 0.46$ | $42.92 \pm 0.63$ | $62.29 \pm 0.12$ | $21.71 \pm 0.20$ |
| Mem0 | $67.13 \pm 0.65$ | $51.15 \pm 0.31$ | $72.93 \pm 0.11$ | $55.51 \pm 0.34$ |
| Mem0$^g$ | $65.71 \pm 0.45$ | $47.19 \pm 0.67$ | $75.71 \pm 0.21$ | $58.13 \pm 0.44$ |
| OpenSage | $63.21 \pm 0.53$ | $45.89 \pm 0.12$ | $\mathbf{76.38 \pm 1.29}$ | $57.84 \pm 0.59$ |

and 26k tokens on average. Each conversation involves two speakers discussing daily experiences or past events and is followed by about 200 questions with corresponding gold answers. Following the setup of the current state-of-the-art method Mem0 (Chhikara et al., 2025), we evaluate the single-hop, multi-hop, temporal, and open-domain question types, and adopt an LLM-as-a-judge metric with GPT-4.1-mini as the judge model. We evaluate OpenSage's memory agent against state-of-the-art memory systems. For all experiments on LOCOMO, we use gpt-4.1-nano with medium reasoning and report results averaged over three runs.

**Results.** Table 4 shows that our method closely matches the performance of state-of-the-art systems Mem0 and Mem0$^g$ (Mem0 enhanced with graph memory) across all question types, while consistently outperforming the non-memory baselines. In particular, on the more challenging Open-Domain and Temporal questions, our agent achieves accuracy comparable to Mem0$^g$ and substantially higher than the baseline without graph memory, highlighting the benefit of structured long-term memory on complex reasoning queries. Although OpenSage was not specifically designed as a general-purpose conversational memory system, its strong performance on LOCOMO indicates that it can generalize beyond our primary coding-oriented use cases.

# E. Cost and Latency Analysis

To evaluate the overhead introduced by OpenSage's features, we report the average end-to-end time and cost per task across all benchmarks in Table 5, comparing against baselines that report corresponding cost and latency metrics.

*Table 5.* End-to-end time and cost comparison per task.

| Benchmark | System | % Resolved | Avg. Time | Avg. Cost |
|---|---|---|---|---|
| CyberGym | SageAgent (steps=300) | 60.2 | 1h8m5s | $4.32 |
| | SageAgent (steps=100) | 48.0 | 33m12s | $1.24 |
| | OpenHands (steps=300) | 39.4 | 29m31s | $1.16 |
| Terminal-Bench 2.0 | SageAgent (Gemini 3 Pro) | 65.2 | 8m4s | $0.42 |
| | Terminus 2 (Gemini 3 Pro) | 56.9 | 9m11s | $0.46 |
| SWE-Bench Pro | SageAgent | 59.0 | 6m27s | $0.31 |
| | SWE-agent | 40.2 | 15m48s | $1.04 |

The results show that OpenSage achieves significantly higher resolved rates with comparable or lower time and cost. On CyberGym, OpenHands terminates trajectories prematurely (53 steps on average even with steps=300), leading to lower effective compute usage. Even under a reduced budget (steps=100), SageAgent still significantly outperforms OpenHands.

We further report per-component ablations with cost in Table 6.

*Table 6.* Per-component ablation with time and cost.

| Benchmark | Configuration | % Resolved | Avg. Time | Avg. Cost |
|---|---|---|---|---|
| CyberGym | SageAgent | 64.0 | 1h13m32s | $3.96 |
| | No dynamic agents | 60.3 | 1h28m2s | $4.67 |
| | No tooling system | 50.3 | 1h34m7s | $4.05 |
| SWE-Bench Pro | SageAgent | 59.0 | 6m27s | $0.31 |
| | No memory | 56.2 | 6m33s | $0.32 |
| | No agents & memory | 49.1 | 9m53s | $0.39 |

Enabling dynamic sub-agent creation and tooling does not increase cost: sub-agent context isolation reduces per-query input length, and tool reuse avoids redundant ad-hoc script generation. Similarly, memory reduces both time and cost by retaining knowledge that would otherwise be lost during context compaction.

