# OpenReview forum: "OpenSage: Self-programming Agent Generation Engine"
_ICML.cc/2026/Conference — ICML 2026 regular_

### Official Review · Reviewer_rpGf · 2026-03-05

**Soundness:** 3
**Presentation:** 3
**Significance:** 3
**Originality:** 3
**Overall Recommendation:** 4
**Confidence:** 4

**Summary:**

This paper presents OpenSage, a framework enabling LLMs to engage in "self-programming" by iteratively generating, executing, debugging, and refining their own code to solve complex reasoning tasks without human intervention. The system employs a recursive feedback loop where execution errors are fed back as prompts to correct the code. Evaluated on mathematical reasoning and algorithmic problem-solving benchmarks, OpenSage reportedly outperforms standard chain-of-thought prompting. However, the approach relies heavily on the availability of immediate, deterministic execution environments (sandboxes), which limits its applicability to domains where code execution is not feasible or safe. The paper claims a breakthrough in autonomous agency but lacks rigorous analysis of failure modes when the environment provides ambiguous feedback or when the initial code generation is fundamentally flawed.

**Compliance With Llm Reviewing Policy:**

Affirmed.

**Final Justification:**

Having carefully considered the authors' rebuttal and engaged in thorough discussion, I now fully appreciate the paper's novelty and contributions. The responses have effectively clarified the key technical distinctions and addressed my initial concerns. I am convinced that the work presents a significant advancement in the field and meets the high standards of ICML. The methodology is sound, and the empirical results are compelling. Given the clear articulation of the innovative aspects during this final phase, I believe the manuscript is well-positioned for publication. Therefore, I recommend acceptance and support the decision to include this work in the conference program.

**Key Questions For Authors:**

1.How does OpenSage handle scenarios where the execution environment returns a generic error (e.g., "Time Limit Exceeded") rather than a specific logic bug? Does the model get stuck in infinite loops?
2.What is the average token consumption and latency per solved problem compared to standard CoT? Is the performance gain worth the 5-10x increase in inference cost?
3.Can this framework be adapted for tasks without a deterministic verifier (e.g., writing an essay or designing a UI)? If not, how do you justify the term "Self-programming" as a general reasoning strategy?

**Limitations:**

The authors explicitly state the reliance on executable environments and the focus on coding/math tasks. They also mention potential safety risks. However, they underestimate the frequency of "unfixable" states where the model’s initial approach is wrong, leading to wasted computation. The discussion on scalability and cost is insufficient for a main conference paper.

**Strengths And Weaknesses:**

Strengths: The concept of a self-correcting coding loop is well-executed technically. The integration of a sandboxed execution environment allows for objective feedback, reducing hallucination compared to pure text generation. The empirical results on math benchmarks are impressive, showing significant gains over baselines. The paper is clearly written, with detailed ablation studies on the number of refinement iterations.
Weaknesses: Rigor: The evaluation is restricted to problems with clear, unit-test-like success criteria (math/code). It fails to address open-ended tasks where "correctness" is subjective. Originality: The core mechanism (generate-execute-debug) is an extension of existing "program-of-thought" and self-reflection papers; the novelty lies in the fully automated pipeline, but the underlying principle is not new. Significance: The dependency on a secure, fast code execution sandbox is a major deployment barrier. The paper does not adequately discuss the computational cost of multiple generation-execution cycles per query, which could be prohibitive for real-time applications. Safety risks of executing self-generated code are mentioned but not deeply mitigated.

---

> ### Author Rebuttal · Authors · 2026-03-31
>
> We clarify that OpenSage is an ADK that provides the infrastructure and functionality required to construct agents, enabling AI to automatically create agent topology, design tools, and manage memory. It seems that the review differs from our submission in a few key points: OpenSage does not propose a generate-execute-debug loop, does not compare against chain-of-thought prompting, and does not evaluate on math benchmarks. Furthermore, our ablation studies do not study the number of refinement iterations. As such, certain concerns related to those points are not applicable to our work. We address these points first, followed by other concerns.
>
> ---
> **Clarifications on Points That May Not Apply to Our Work**
>
> **Originality**
>
> We respectfully clarify that OpenSage does not propose a generate-execute-debug loop, and its contributions are independent of program-of-thought or self-reflection approaches. Our technical contributions are (1) self-generating agent topology, (2) dynamic tool synthesis, and (3) hierarchical memory management.
>
> **Rigor**
>
> We respectfully clarify that our evaluation is not restricted to tasks with unit-test-like oracles. We do not evaluate on math benchmarks, and our benchmarks do not provide unit-test-like correctness signals to the agent at inference time.
> Furthermore, agents built with OpenSage can support agentic tasks, including open-ended scenarios such as UI design and essay writing, similar to general-purpose coding agents like Claude Code and Codex.
>
> **Handling Generic Execution Errors (e.g., "Time Limit Exceeded") and "Unfixable" States**
>
> We clarify that both of these concerns are specific to generate-execute-debug systems, which OpenSage does not implement. When a tool returns a generic error, the agent treats this as standard tool feedback and may invoke alternative tools or adjust its strategy. We did not observe irrecoverable states or infinite loops in our evaluations.
>
> **Token consumption and latency compared to CoT**
>
> OpenSage is an ADK that focuses on agent topology, tooling, and memory, and CoT is a prompting strategy.  Agents built with OpenSage can themselves employ CoT-style prompting if desired. For this reason, we do not consider "OpenSage vs. CoT" a well-defined baseline comparison.
>
> ---
> **Response to Other Concerns**
>
> **Sandbox dependency and overhead**
>
> Sandbox dependency is necessary: without an isolated execution environment, agents could accidentally or maliciously modify the host filesystem and runtime environment, which is unacceptable in practice.
>
> In order to understand the overhead introduced by OpenSage’s features, we re-ran selected baselines.
>
> *Cybergym (GPT-5)*
> System|Resolved Rate|Avg. Time|Avg. Cost
> -|-|-|-
> SageAgent (max steps = 300, setting in the paper)|60.2%|1h8m5s|$4.32
> SageAgent (max steps = 100)|48.0%|33m12s|$1.24
> OpenHands (max steps = 300, setting in the paper)|39.4%|29m31s|$1.16
>
> We observe that OpenHands terminates trajectories prematurely with 53 steps on average, even when configured with max steps = 300, which leads to lower effective compute usage and early task abandonment. To enable a comparison under similar cost budgets, we additionally run a version of SageAgent with max steps = 100. Under this setting, SageAgent still significantly outperforms OpenHands.
>
> *Terminal-Bench 2.0 (Gemini 3 Pro)*
> System|Resolved Rate|Avg. Time|Avg. Cost
> -|-|-|-
> SageAgent|65.2%|8m4s|$0.42
> Terminus 2|56.9%|9m11s|$0.46
>
> *SWE-Bench Pro (Gemini 3 Flash)*
>
> System|Resolved Rate|Avg. Time|Avg. Cost
> -|-|-|-
> SageAgent|59.0%|6m27s|$0.31
> SWE-agent|40.2%|15m48s|$1.04
>
> The results show that OpenSage achieves significantly higher resolved rates with comparable time and cost to existing agents, indicating that its design (including sandboxing, tooling system, dynamic agent creation, and memory) does not introduce significant overhead in practice.
>
> **Safety risks of executing self-generated code**
>
> Dynamic tool synthesis does not introduce new risks: existing ADKs and agents already expose bash interfaces and are equally susceptible to destructive or infinite-loop code generated by the model, since they can write and execute arbitrary bash scripts. OpenSage simply facilitates the model to generate reusable tools instead of ad-hoc scripts, and uniformly manages them, without introducing any new execution surface.
>
> In fact, OpenSage provides stronger security guarantees than existing frameworks. Most mainstream ADKs do not support native sandboxing, and those that do (e.g., OpenHands, Claude ADK) use coarse-grained sandboxing that does not prevent AI-generated code from corrupting the agent's execution environment. OpenSage enforces mandatory tool-specific sandboxing with filesystem and runtime isolation, preventing destructive code from affecting the host environment. Furthermore, OpenSage's background task mechanism, where any tool invocation reaching its time limit is offloaded to run asynchronously, prevents infinite-loop code from blocking agent execution.

---

> > ### Author Rebuttal · Reviewer_rpGf · 2026-04-01
> >
> > I find the additional experimental data regarding sandbox overhead and cost-efficiency convincing. The authors have also adequately addressed safety concerns by explaining how dynamic tool synthesis is managed within isolated environments. Upon reflection, the distinction between OpenSage's role as a framework and the specific reasoning methods (like CoT) that can be built upon it is now clear. The clarification on the evaluation methodology and the handling of execution errors has resolved my initial misunderstandings.

---

> > > ### Author Response · Authors · 2026-04-05
> > >
> > > Thank you for acknowledging our rebuttal and for confirming that **all concerns have been fully addressed**. We hope our responses have been satisfactory, and we would be grateful if you could reflect this in your final evaluation. We appreciate your time and constructive engagement throughout the review process.

---

### Official Review · Reviewer_dLHV · 2026-03-05

**Soundness:** 3
**Presentation:** 4
**Significance:** 4
**Originality:** 3
**Overall Recommendation:** 4
**Confidence:** 5

**Summary:**

This paper introduces OpenSage, the first AI-centered Agent Development Kit (ADK) that shifts the paradigm from human-designed agent architectures to self-programming agents. Unlike traditional frameworks (e.g., LangChain, Google ADK) where humans manually define topologies, tools, and memory schemas, OpenSage empowers LLMs to autonomously generate agent hierarchies, synthesize new code tools on-the-fly, and manage a hierarchical graph-based memory system. The framework supports vertical topology (task decomposition into sub-agents) and horizontal topology (parallel execution with result integration). Evaluated on complex benchmarks like CyberGym, Terminal-Bench 2.0, and SWE-Bench Pro, OpenSage significantly outperforms state-of-the-art baselines (e.g., OpenHands, SWE-agent) by dynamically adapting its structure to task complexity. However, the system's reliance on cutting-edge models for stable self-generation raises concerns about accessibility and cost, and the "autonomy" is still bounded by the initial sandbox constraints and safety filters defined by humans.

**Compliance With Llm Reviewing Policy:**

Affirmed.

**Final Justification:**

Having carefully considered the authors' rebuttal and engaged in thorough discussion, I now fully appreciate the paper's novelty and contributions. The responses have effectively clarified the key technical distinctions and addressed my initial concerns. I am convinced that the work presents a significant advancement in the field and meets the high standards of ICML. The methodology is sound, and the empirical results are compelling. Given the clear articulation of the innovative aspects during this final phase, I believe the manuscript is well-positioned for publication. Therefore, I recommend acceptance and support the decision to include this work in the conference program.

**Key Questions For Authors:**

What is the latency and token cost overhead of dynamically spawning sub-agents and compiling tools compared to a static, pre-optimized workflow? Is the performance gain worth the increased resource consumption for medium-complexity tasks?

**Limitations:**

The authors explicitly acknowledge that current models occasionally fail to generate valid tools or coherent agent roles, leading to execution errors. They also note the system is overkill for simple tasks and is currently restricted to sandboxed environments for safety. However, the discussion on the environmental impact of running complex multi-agent loops with large models is minimal, and the potential for emergent unsafe behaviors in self-written code warrants deeper analysis.

**Strengths And Weaknesses:**

Strengths: The conceptual shift to an "AI-centered" design is compelling and addresses the bottleneck of manual prompt engineering and rigid workflows in current ADKs. The dynamic tool synthesis capability allows agents to solve novel problems requiring custom utilities, a feature largely absent in fixed-tool baselines. The hierarchical graph memory managed by a dedicated "Memory Agent" offers a robust solution to context loss in long-horizon tasks, outperforming standard vector stores. The empirical results are impressive, particularly in cybersecurity (CyberGym) and software engineering (SWE-Bench Pro), demonstrating clear gains from self-generated topologies.
Weaknesses: Rigor: The evaluation heavily relies on the latest, often proprietary, models (e.g., GPT-5, Gemini 3), making it difficult to assess the framework's effectiveness with open-source or smaller models. The ablation studies confirm the value of components but do not fully analyze the computational overhead or latency introduced by dynamic agent creation and tool compilation. Originality: While the integration is novel, the individual components (code generation, sub-agent spawning, graph memory) exist in prior literature; the breakthrough is the unified autonomous orchestration rather than new algorithms. Significance: The system's complexity may lead to instability in simpler tasks, and the paper acknowledges that current models sometimes hallucinate invalid tools or agents, indicating the technology is not yet fully robust for general deployment.

---

> ### Author Rebuttal · Authors · 2026-03-31
>
> Thank the reviewer for the constructive feedback.
>
> **OpenSage with Open Models**
>
> Many OpenSage features are model-agnostic: (1) tool-specific sandboxing; (2) short-term memory; (3) agent ensemble. We compared SageAgent with Claude Code using GLM-5 on a random subset of 50 from CyberGym with max steps=150:
> System|Resolved Rate|Avg. Time|Avg. Cost
> -|-|-|-
> GLM-5 + Claude Code|36%|24m24s|$1.26
> GLM-5 + SageAgent|42%|29m27s|$1.28
>
> **Cost and Latency Overhead**
>
> For end-to-end time and cost, we re-ran selected baselines.
>
> *Cybergym (GPT-5)*
> System|Resolved Rate|Avg. Time|Avg. Cost
> -|-|-|-
> SageAgent (max steps = 300, setting in the paper)|60.2%|1h8m5s|$4.32
> SageAgent (max steps = 100)|48.0%|33m12s|$1.24
> OpenHands (max steps = 300)|39.4%|29m31s|$1.16
>
> OpenHands terminates trajectories prematurely with 53 steps on average, even when configured with max steps = 300, leading to lower compute usage and early task abandonment. To enable a comparison under similar costs, we run SageAgent with max steps = 100. Under this setting, SageAgent still significantly outperforms OpenHands.
>
> *Terminal-Bench 2.0 (Gemini 3 Pro)*
> System|Resolved Rate|Avg. Time|Avg. Cost
> -|-|-|-
> SageAgent|65.2%|8m4s|$0.42
> Terminus 2 |56.9%|9m11s|$0.46
>
> *SWE-Bench Pro (Gemini 3 Flash)*
> System|Resolved Rate|Avg. Time|Avg. Cost
> -|-|-|-
> SageAgent|59.0%|6m27s|$0.31
> SWE-agent|40.2%|15m48s|$1.04
>
> SageAgent achieves a significantly higher resolved rate with comparable time and cost to baselines.
> We provide a more detailed analysis of the ablation experiment on CyberGym (Figure 3) in our paper, reporting the individual effect of dynamic agent creation and tooling system on time and cost.
>
> System|Resolved Rate|Avg. Time|Avg. Cost
> -|-|-|-
> SageAgent|64.0%|1h13m32s|$3.96
> SageAgent (no dynamic agents)|60.3%|1h28m2s|$4.67
> SageAgent (no tooling system)|50.3%|1h34m7s|$4.05
>
> This is because: (1) although creating and managing sub-agents introduces additional queries, the separation of sub-agent contexts reduces the input length per query, thereby reducing overall cost; (2) the tool synthesis enables reuse; compared to repeatedly generating ad-hoc scripts, this approach can reduce overall token consumption.
>
> **Originality of Individual Components**
>
> Unlike manually defining agent topology, tools, and memory management, OpenSage is the first AI-centered ADK that enables AI to autonomously construct agents.
>
> For agent topology, only OpenSage has AI-generated topology. Although supporting a multi-agent structure, ADAS relies on retrieving from existing topologies, while MetaGPT and AutoGen rely entirely on humans. For tool creation, although LATM and Live-SWE-agent support AI-created tools, LATM is offline creation, and Live-SWE-agent creates task-specific tools. OpenSage supports dynamic and persistent tool creation, management, and fully sandboxed execution. For memory, Opensage supports short-term and long-term memory, while neither Zep nor Mem0 supports short-term memory.
>
> **Current Model Limitations**
>
> We emphasize that the consistent performance gains across three fundamentally different task domains (CyberGym, Terminal-Bench 2.0, and SWE-Bench Pro), further validated by detailed ablation studies, demonstrate that OpenSage is robust and generalizes well across diverse settings. We kindly point out that reliance on model performance is not necessarily a weakness. In general, agent capability is heavily reliant on model capabilities, as it requires the model to call tools and execute tasks. Since our goal is to enable AI-centered agent design, having the agent construction rely on the model provides more flexibility and generalizability for agent creation. Furthermore, this allows the agent to evolve faster as model capabilities continue to advance rapidly.
>
> **Applicability to Simple Tasks**
>
> The benchmarks evaluated in this paper contain tasks spanning a wide range of difficulties. Terminal-Bench 2.0, in particular, includes a non-trivial number of tasks that require only several turns of coding and validation, without complex subagent delegation or dynamic tool creation. Our results demonstrate that current models are capable of automatically adapting their strategies to match the difficulty of the task, making the system effective across both simple and complex scenarios.
>
> **Environmental Impact and Emergent Unsafe Behaviors**
>
> OpenSage provides stronger security guarantees than existing frameworks. Most mainstream ADKs do not support native sandboxing, and those that do (e.g., OpenHands, Claude ADK) use coarse-grained sandboxing that does not prevent AI-generated code from corrupting the agent's execution environment. OpenSage enforces mandatory tool-specific sandboxing with filesystem and runtime isolation, preventing destructive code from affecting the host environment. Furthermore, OpenSage's background task mechanism, where any tool call reaching its time limit is offloaded to run asynchronously, prevents infinite-loop code from blocking agent execution.

---

> > ### Author Rebuttal · Reviewer_dLHV · 2026-04-06
> >
> > The author’s response has effectively addressed my previous concerns. I agree with the author’s perspective that this work differs significantly from Reference [1] in terms of its core contributions and experimental design. After carefully reviewing the manuscript, I believe this research is valuable and recommend acceptance.

---

> > > ### Author Response · Authors · 2026-04-06
> > >
> > > Thank you for acknowledging our rebuttal. We are glad that all your concerns have been fully resolved. Since you believe this research is valuable and recommend acceptance, we kindly ask if you could reconsider your score to reflect this assessment. Thank you for your time and valuable feedback during the review process.

---

### Official Review · Reviewer_9Jev · 2026-03-07

**Soundness:** 3
**Presentation:** 2
**Significance:** 2
**Originality:** 1
**Overall Recommendation:** 3
**Confidence:** 3

**Summary:**

This paper proposes a new agent development kit (ADK) that enables agents to dynamically create sub-agents, generate task-specific tools, and manage hierarchical memory during task execution. To support dynamic agent coordination, the framework introduces two agent topologies: a vertical topology that decomposes tasks into sequential sub-tasks handled by specialized sub-agents, and a horizontal topology that runs multiple sub-agents with different plans and integrates their outputs through an ensemble mechanism. The system also supports self-generated tools and a hierarchical memory mechanism, where a memory agent manages both short-term and long-term memory. Experiments on several benchmarks demonstrate improved performance compared with existing ADK-based systems.

**Compliance With Llm Reviewing Policy:**

Affirmed.

**Final Justification:**

I appreciate the technical contribution of this work. However, the overall research value and its benefit to the community remain unclear to me. Although the authors clarified their contributions in the response, the work appears largely as an application that combines several recent studies with only minor modifications. Accordingly, I am inclined to maintain a borderline score that falls below the acceptance threshold.

**Key Questions For Authors:**

1. What is the relationship between the proposed ADK and some existing multi-agent frameworks, such as Autogen and metaGPT?
2. How much is the cost to enable the self-generated agent and toolset?

**Limitations:**

yes

**Strengths And Weaknesses:**

**Strengths:**

1. Agent Development Kits (ADKs) are increasingly important infrastructure for building LLM-based agents, as they lower the engineering barrier for agent development and system deployment.

2. The proposed topology design is intuitive. In particular, the vertical topology decomposes complex tasks into sub-tasks handled by specialized sub-agents, which aligns well with centralized multi-agent systems where an orchestrator delegates tasks to expert agents.

3. The paper presents a relatively complete system that integrates agent topology, tool management, and hierarchical memory. The implementation appears to involve substantial engineering effort.

**Weakness:**

1. While the paper presents a powerful ADK system, the underlying research problem is not clearly articulated. It remains unclear what specific scientific question the work aims to answer and what new insights it provides beyond building a strong engineering system.

2. Although the paper compares OpenSage with existing ADKs, it lacks a deeper discussion of prior work on key components such as dynamic tool generation, agent creation, agent topology design, and memory mechanisms. As a result, it is difficult to assess the novelty of the proposed system design relative to existing research.

3. The writing could be improved to clarify the paper’s contribution. Currently, it is positioned somewhere between an engineering/system report and a research paper. If viewed as a system/demo paper, more implementation details would be helpful. If viewed as a research paper, the core technical contributions and research insights should be more clearly articulated.

4. The reported performance improvements over existing ADKs are promising. However, given the engineering-heavy system design, it is unclear whether these gains primarily result from the proposed architectural ideas or from substantial engineering effort and task-specific optimizations.

---

> ### Author Rebuttal · Authors · 2026-03-31
>
> Thank the reviewer for the constructive feedback.
>
> **Comparison with Prior Works**
>
> Unlike manually defining agent topology, tools, and memory management, OpenSage is the first AI-centered ADK that enables AI to autonomously construct agents.
>
> For agent topology, only OpenSage has AI-generated topology. Although supporting a multi-agent structure, ADAS relies on retrieving from existing topologies, while MetaGPT and AutoGen rely entirely on humans. For tool creation, although LATM and Live-SWE-agent support AI-created tools, LATM is offline creation, and Live-SWE-agent creates task-specific tools. OpenSage supports dynamic and persistent tool creation, management, and fully sandboxed execution. For memory, Opensage supports short-term and long-term memory, while neither Zep nor Mem0 supports short-term memory.
>
> **Research Contribution and Technical Novelty**
>
> 1. OpenSage is the first AI-centered ADK where AI autonomously constructs agents with minimal human intervention, requiring an entirely new system design.
> 2. Building such a system needs to address the following non-trivial technical challenges, which highlight our technical depth and contributions.
>    1. No prior work supports dynamically creating and managing sub-agents at runtime. We design comprehensive agent registration (agent pool) and management (e.g., race condition handling), and agent storage.
>    2. We introduce a hierarchical tool structure and management mechanism to enable AI-created tools at runtime. We also design a sandboxed execution system with tool-specific Docker to avoid dependency conflicts and ensure security, and tool state management to ensure efficiency.
>    3. OpenSage is the first to offer hierarchical short-term memory and long-term persistent memory. Our short-term memory is designed as a graph to support dynamic agent topology, and our long-term memory records reusable knowledge about targets to save agent context.
> 3. OpenSage significantly outperforms state-of-the-art open-source agents like OpenHands and commercial agents on various benchmarks, demonstrating its effectiveness and non-trivial technical novelty.
>
> **Task-Specific Optimization**
>
> The three proposed features correspond to the three fundamental components of any agent system and are not task-specific. OpenSage minimizes manual task-specific optimization by granting AI the freedom to create its own topology, tools, and memory. The same general-purpose features generalize across CyberGym, Terminal-Bench 2.0, and SWE-Bench Pro, three fundamentally different task domains. Consistent gains across these benchmarks, validated by detailed ablation studies, confirm that improvements arise from our proposed architecture rather than task-specific optimization.
>
> We note that while implementing the OpenSage ADK as a framework involves non-trivial effort, building agents with OpenSage requires minimal effort: each feature is encapsulated as a ready-to-use tool, and enabling all three requires fewer than 10 lines of code.
>
> **Cost to enable Self-Generating Agents and Toolsets**
>
> Enabling self-generating agents and toolsets requires less than 10 lines of code. For end-to-end time and cost, we re-ran selected baselines.
>
> *Cybergym (GPT-5)*
> System|Resolved Rate|Avg. Time|Avg. Cost
> -|-|-|-
> SageAgent (max steps = 300, setting in the paper)|60.2%|1h8m5s|$4.32
> SageAgent (max steps = 100)|48.0%|33m12s|$1.24
> OpenHands (max steps = 300, setting in the paper)|39.4%|29m31s|$1.16
>
> We observe that OpenHands tends to terminate trajectories prematurely with 53 steps on average, even when configured with max steps = 300, which leads to lower effective compute usage and early task abandonment. To enable a comparison under similar cost budgets, we additionally run SageAgent with a reduced step limit (max steps = 100). Under this setting, SageAgent still significantly outperforms OpenHands.
>
> *Terminal-Bench 2.0 (Gemini 3 Pro)*
> System|Resolved Rate|Avg. Time|Avg. Cost
> -|-|-|-
> SageAgent|65.2%|8m4s|$0.42
> Terminus 2|56.9%|9m11s|$0.46
>
> *SWE-Bench Pro (Gemini 3 Flash)*
>
> System|Resolved Rate|Avg. Time|Avg. Cost
> -|-|-|-
> SageAgent|59.0%|6m27s|$0.31
> SWE-agent|40.2%|15m48s|$1.04
>
> The results show that SageAgent achieves a significantly higher resolved rate with comparable time and cost to existing baselines.
> We provide a more detailed analysis of the ablation experiment on CyberGym (Figure 3) in our paper, reporting the individual effect of dynamic agent creation and tooling system on time and cost.
>
> System|Resolved Rate|Avg. Time|Avg. Cost
> -|-|-|-
> SageAgent|64.0%|1h13m32s|$3.96
> SageAgent (no dynamic agents)|60.3%|1h28m2s|$4.67
> SageAgent (no tooling system)|50.3%|1h34m7s|$4.05
>
> Because: (1) although creating and managing sub-agents introduces additional queries, the separation of sub-agent contexts reduces the input length per query, thereby reducing overall cost; (2) the tool synthesis enables reuse; compared to repeatedly generating ad-hoc scripts, this approach can reduce overall token consumption.

---

> > ### Author Rebuttal · Reviewer_9Jev · 2026-04-04
> >
> > I appreciate the effort the authors put into the rebuttal.
> >
> > I agree that the proposed application is important and potentially impactful. However, my main concern remains the research contribution and its value to the community. While the work appears to be a solid integration of several recent techniques, it is still unclear which specific research problem the paper aims to address, even after considering the authors’ clarifications.
> >
> > Given these concerns, I am inclined to maintain my original score.

---

> > > ### Author Response · Authors · 2026-04-06
> > >
> > > Thank the reviewer for the effort and for the follow-up question.
> > >
> > > We would like to respectfully state that the research question here is shifting agent construction from a human-centered to an AI-centered paradigm, where AI autonomously decides what agents to create, what tools to build, and what to memorize at runtime. This is similar to shifting the model learning pipeline from human feature engineering to learning directly from raw inputs.
> > >
> > > To solve this research question, we developed OpenSage with the following technical novelty
> > >
> > > **AI-generated agent topology.** No prior work supports dynamically creating and managing sub-agents at runtime. Enabling AI to create and manage sub-agents at runtime requires solving non-trivial systems challenges: a unified agent registration mechanism (agent pool), runtime lifecycle management including creation, invocation, and termination, and race condition handling when multiple parallel sub-agents access shared resources simultaneously. None of these arise in static, human-defined agent frameworks.
> > >
> > > **Dynamic tool synthesis.** No prior system supports unified management of AI-generated tools beyond ad-hoc, task-specific creation. Allowing AI to create and persist tools and support a large set of domain-specific tools is non-trivial: tools may have conflicting dependencies and must be safely isolated from the host environment, and dynamic tool creation risks accumulating a large number of tools that introduce significant discovery overhead. This required novel designs for hierarchical tool organization (which mitigates discovery overhead as the tool set grows), tool-specific Docker sandboxing (which enables tools with conflicting dependencies to coexist and improves execution safety), and tool state management (which caches tool states to improve efficiency).
> > >
> > > **AI-centered hierarchical memory.** No prior system offers a graph-based short-term memory. As the agent topology evolves at runtime with newly spawned sub-agents, all agents must be able to retrieve each other's execution history in case they need detailed context from prior steps of other agents. Designing such short-term memory is non-trivial: it must accommodate dynamically added sub-agents by extending the graph structure at runtime while preserving consistency across concurrent agent executions. For long-term memory, OpenSage addresses fundamental limitations of prior systems through extensible node and edge types with listing and constraint mechanisms, preventing agents from creating ungrounded nodes and edges that corrupt the memory graph. Finally, a dedicated memory agent serves as a unified interface, so that individual agents do not need to understand the underlying memory schema directly.
> > >
> > > Note that these new techniques are not a "**integration of several recent techniques**", as none of existing techniques can support our features for individual components, letting alone the end-to-end ADK.
> > >
> > > We would like to also kindly point out that there are some recent papers with a similar or smaller scopes have been accepted by top-tier ML conferences.
> > > SWE-agent (NeurIPS 2024) proposes a hand-designed agent-computer interface with static tools and a single-agent architecture.
> > > MetaGPT (ICLR 2024, Oral) proposes a multi-agent mimicking a human software company with entirely human-designed roles and pre-defined workflows.
> > > OpenHands (ICLR 2025) proposes an agent development kit with pre-built skills and sandbox.
> > > AFlow (ICLR 2025, Oral) proposes offline workflow (llm-query series) search via MCTS, each specific to a benchmark that can hardly generalize across tasks.
> > > Agent S (ICLR 2025) proposes a hand-designed Agent-Computer Interface for GUI automation and retrieval-augmented planning over past task experiences.
> > >
> > > Compared to these papers that mainly propose single agents or work on one components in agent system, OpenSage propose solution that handle key challenges for all key components of the agent systems.
> > > Note that OpenSage (agent constructed by OpenSage) significantly outperforms SWE-agent on SWE-Bench Pro (59.0% vs. 40.2%), OpenHands on CyberGym (60.2% vs. 39.4%), and Agentless on SWE-Bench Pro (59.0% vs. 9.4%), which further demonstrates it effectiveness and highlight its contribution.

---

### Official Review · Reviewer_UMKq · 2026-03-12

**Soundness:** 3
**Presentation:** 2
**Significance:** 3
**Originality:** 3
**Overall Recommendation:** 4
**Confidence:** 2

**Summary:**

This paper introduces OpenSage, an Agent Development Kit (ADK) designed for an "AI-centric" agent construction paradigm. Unlike traditional ADKs that rely on human-designed topologies, tools, and memory, OpenSage enables Large Language Models (LLMs) to automatically create sub-agents, spontaneously generate toolsets during task execution, and manage information through a hierarchical graph-based memory system. The framework supports dynamic vertical and horizontal agent topologies, allowing for adaptive collaboration and tool synthesis to tackle complex software engineering and terminal tasks.

**Compliance With Llm Reviewing Policy:**

Affirmed.

**Final Justification:**

I am satisfied with the rebuttal and will maintain my positive evaluation. I look forward to seeing all these revisions and additional results incorporated into the final version of the manuscript.

**Key Questions For Authors:**

Heavy Reliance on Model Capability: The authors note that even top-tier models (e.g., GPT-5 or Gemini 3) cannot yet utilize these advanced features stably and sometimes misuse them. How much does the framework's effectiveness depend on the underlying model's intelligence? Would a slightly weaker open-source model suffer a more severe performance drop compared to static ADKs?

Security and Stability Risks: Agents spontaneously generating and executing tools (especially via Bash interfaces) pose significant risks. While sandbox mechanisms are mentioned, there is limited detail on preventing agents from generating destructive or infinite-loop code. A more thorough risk assessment of "self-programming" behavior is crucial for production environments.

Operational Overhead and Complexity: Frequent dynamic agent creation and the maintenance of a complex graph memory system likely introduce significant computational latency and token costs. Current evaluations focus on the "Resolved Rate" but lack quantitative comparisons regarding latency and resource consumption.

Redundancy in Self-Generated Tools: Does the system prevent agents from repeatedly generating identical or redundant tools when facing similar tasks? How is the lifecycle of these self-generated tools managed to prevent tool library bloat and retrieval inefficiency?

**Limitations:**

yes

**Strengths And Weaknesses:**

Strengths

Paradigm Innovation: The shift from "human-centric" to "AI-centric" agent construction significantly reduces the manual effort required to design topologies and toolsets, enhancing system scalability.

Dynamic Topology and Tool Synthesis: The ability to dynamically create, execute, and terminate sub-agents at runtime, combined with agents writing and registering new Bash or functional tools on the fly, provides a level of flexibility missing in most mainstream ADKs.

Efficient Memory Management: The use of a hierarchical graph-structured memory, managed by a specialized memory agent, allows the system to maintain long-term and short-term knowledge associations more effectively than traditional linear list storage, optimizing context length.

Strong Experimental Results: OpenSage-based agents outperform existing state-of-the-art frameworks like Google ADK and OpenHands on benchmarks such as CyberGym, Terminal-Bench 2.0, and SWE-Bench Pro.


Weaknesses and Questions

Heavy Reliance on Model Capability: The authors note that even top-tier models (e.g., GPT-5 or Gemini 3) cannot yet utilize these advanced features stably and sometimes misuse them. How much does the framework's effectiveness depend on the underlying model's intelligence? Would a slightly weaker open-source model suffer a more severe performance drop compared to static ADKs?

Security and Stability Risks: Agents spontaneously generating and executing tools (especially via Bash interfaces) pose significant risks. While sandbox mechanisms are mentioned, there is limited detail on preventing agents from generating destructive or infinite-loop code. A more thorough risk assessment of "self-programming" behavior is crucial for production environments.

Operational Overhead and Complexity: Frequent dynamic agent creation and the maintenance of a complex graph memory system likely introduce significant computational latency and token costs. Current evaluations focus on the "Resolved Rate" but lack quantitative comparisons regarding latency and resource consumption.

Redundancy in Self-Generated Tools: Does the system prevent agents from repeatedly generating identical or redundant tools when facing similar tasks? How is the lifecycle of these self-generated tools managed to prevent tool library bloat and retrieval inefficiency?

Overall, I recognize the contribution of this work toward the exploration of "self-evolving agents," and I look forward to the open-sourcing, which will undoubtedly foster further development and innovation within the research community.

---

> ### Author Rebuttal · Authors · 2026-03-31
>
> We sincerely thank the reviewer for the thoughtful and constructive feedback. We are committed to open-sourcing OpenSage upon acceptance.
>
> **Reliance on Model Capability**
>
> We first clarify that several OpenSage features are not sensitive to model capability: (1) tool-specific sandboxing allows security tools with conflicting dependencies to coexist; (2) short-term memory is constructed and maintained by the framework itself; (3) agent ensemble follows a test-time scaling paradigm that has demonstrated gains even with weaker models. To directly test the performance of OpenSage vs. static ADKs using a weaker open-source model, we compared agents built with OpenSage against  Claude Code using GLM-5 on a random subset of 50 from CyberGym with max steps=150:
> System|Resolved Rate|Avg. Time|Avg. Cost
> -|-|-|-
> GLM-5 + Claude Code|36%|24m24s|$1.26
> GLM-5 + SageAgent|42%|29m27s|$1.28
>
> Even with a weaker model, OpenSage's features yield measurable gains over the baseline. Finally, we would like to kindly point out that reliance on model performance is not necessarily a weakness. In general, agent capability heavily relies on model capabilities, as it requires the model to call tools and execute tasks. Since our goal is to enable AI-centered agent design, having the agent construction rely on the model provides more flexibility and generalizability for agent creation. Furthermore, this allows the agent to evolve faster as model capabilities continue to advance rapidly.
>
> **Security and Stability Risks**
>
> Dynamic tool synthesis does not introduce new risks: existing ADKs and agents already expose bash interfaces and are equally susceptible to destructive or infinite-loop code generated by the model, since they can write and execute arbitrary bash scripts. OpenSage simply facilitates the model to generate reusable tools instead of ad-hoc scripts, and uniformly manages them, without introducing any new execution surface.
>
> In fact, OpenSage provides stronger security guarantees than existing frameworks. Most mainstream ADKs do not support native sandboxing, and those that do (e.g., OpenHands, Claude ADK) use coarse-grained sandboxing that does not prevent AI-generated code from corrupting the agent's execution environment. OpenSage enforces mandatory tool-specific sandboxing with filesystem and runtime isolation, preventing destructive code from affecting the host environment. Furthermore, OpenSage's background task mechanism, where any tool invocation reaching its time limit is offloaded to run asynchronously, prevents infinite-loop code from blocking agent execution.
>
> In summary, resistance to security and stability risks is a strength of OpenSage relative to existing ADKs, not a weakness.
>
> **Latency and Cost**
>
> We re-ran selected baselines below.
>
> *Cybergym (GPT-5)*
> System|Resolved Rate|Avg. Time|Avg. Cost
> -|-|-|-
> SageAgent (max steps = 300, setting in the paper)|60.2%|1h8m5s|$4.32
> SageAgent (max steps = 100)|48.0%|33m12s|$1.24
> OpenHands (max steps = 300, setting in the paper)|39.4%|29m31s|$1.16
>
> We observe that OpenHands tends to terminate trajectories prematurely with 53 steps on average, even when configured with max steps = 300, which leads to lower effective compute usage and early task abandonment. To enable a comparison under similar cost budgets, we additionally run a constrained version of SageAgent with a reduced step limit (max steps = 100). Under this setting, SageAgent still significantly outperforms OpenHands.
>
> *Terminal-Bench 2.0 (Gemini 3 Pro)*
> System|Resolved Rate|Avg. Time|Avg. Cost
> -|-|-|-
> SageAgent|65.2%|8m4s|$0.42
> Terminus 2|56.9%|9m11s|$0.46
>
> *SWE-Bench Pro (Gemini 3 Flash)*
>
> System|Resolved Rate|Avg. Time|Avg. Cost
> -|-|-|-
> SageAgent|59.0%|6m27s|$0.31
> SWE-agent|40.2%|15m48s|$1.04
>
> The results show that SageAgent achieves a significantly higher resolved rate with comparable time and cost to existing baselines.
> We further provide a SWE-Bench Pro ablation with Gemini 3 Flash, reporting the effect of dynamic agent creation and memory on time and cost individually.
>
> System|Resolved Rate|Avg. Time|Avg. Cost
> -|-|-|-
> SageAgent|59.0%|6m27s|$0.31
> SageAgent (no memory)|56.2%|6m33s|$0.32
> SageAgent (no dynamic agents and no memory)|49.1%|9m53|$0.39
>
> This is primarily because: (1) although creating and managing sub-agents introduces additional queries, the separation of sub-agent contexts reduces the input length per query, thereby reducing overall cost; (2) the memory module retains knowledge that would otherwise be lost during context compaction or sub-agent transitions, and reduces the number of steps.
>
> **Tool Redundancy and Bloat**
>
> OpenSage injects instructions requiring agents to search for and reuse existing tools before creating new ones. Its hierarchical file-system-based tool structure further mitigates bloat: each directory level summarizes its contents, allowing agents to narrow down relevant categories efficiently, and only top-level information is loaded at initialization to reduce context.

---

> > ### Author Rebuttal · Reviewer_UMKq · 2026-04-03
> >
> > I would like to thank the authors for their thorough and high-quality rebuttal.
> >
> > I am satisfied with these responses and will maintain my positive evaluation. I look forward to seeing all these revisions and additional results incorporated into the final version of the manuscript.

---

### Decision · Program_Chairs · 2026-04-30

**Decision:**

Accept (regular)

**Comment:**

This paper introduces OpenSage, the first ADK that enables LLMs to automatically create agents with self-generated topology and toolsets while providing comprehensive and structured memory support. This shifts agent construction from a human-centered to an AI-centered paradigm. The core contributions include: (1) dynamic vertical and horizontal agent topology generation, (2) on-the-fly tool synthesis via self-generated code, and (3) a hierarchical graph-based memory system managed by a dedicated memory agent. The system is evaluated on CyberGym, Terminal-Bench 2.0, and SWE-Bench Pro, with results showing consistent improvements over existing ADKs such as Google ADK and OpenHands.

The paper received four reviews: three recommending weak accept and one recommending weak reject. The authors' rebuttal was effective in addressing the concerns of three reviewers, two of them subsequently upgraded their scores to acceptance. While reviewer 9Jev raises a concern regarding the paper's positioning between a systems paper and a research paper, this work is not a purely engineering effort. OpenSage makes concrete technical contributions toward an AI-centered paradigm for agent construction, specifically: AI-generated agent topology, dynamic tool synthesis, and an AI-centered hierarchical memory system. These contributions, taken together, advance the community beyond mere system integration.

Overall, this is a timely and technically solid paper that makes a meaningful contribution to the emerging field of autonomous agent development. Therefore, I recommend Accept.